# The Recovery of Phosphorus from Acidic Ultra-High Phosphorous Wastewater by the Struvite Crystallization

**Qiang Li [1,†], Song Wang [2,3,†], Lifang Wang [1], Li Zhang [3,*], Xiaohui Wan [3] and Zhiguo Sun [3,*]** 

1   School of Management, Northwestern Polytechnical University, 127 West Youxi Road, Xian 710072, China; failureend@163.com (Q.L.); lifang@nwpu.edu.cn (L.W.)
2   Shangtex Architectural Design Research Institute Co., Ltd., Shanghai 200060, China; 13122337565@163.com
3   Research Center of Resource Recycling Science and Engineering, School of Environmental and Materials Engineering, Shanghai Polytechnic University, Shanghai 201209, China; wxhwanxiaohui@163.com
*   Correspondence: zhangli@sspu.edu.cn (L.Z.); zgsun@sspu.edu.cn (Z.S.); Tel.: +86-15021711269 or +86-021-50211210 (L.Z.)
†   These authors contribute equally to this work.

**Abstract:** Phosphorus recovery from industrial wastewater has attracted considerable interest. In this study, struvite crystallization method has been used for treatment of high phosphorus wastewater. The new combination agents of $Mg_5(CO_3)_4(OH)_2·4H_2O$ and $NH_4Cl$ were used as the precipitant. The effects of initial pH, n(Mg):n(P), n(N):n(P), and reaction time on the removal of total phosphorus (TP) in wastewater were investigated. The results showed that under the condition of initial pH = 4, Mg:N:P = 1.2:1.1:1, reaction time for 30 min, and static storage for 20 min, the residual amount of TP in wastewater was 2.98 mg /L, and the removal rate of TP reached 99.99%. The mass fraction of $P_2O_5$ in the generated sediment reached 25.22%, equivalent to high grade phosphate ore and slow-release fertilizer, so as to realize the recycling and utilization of phosphorus in ultra-high phosphorous wastewater. This work will have practical application potential in treatment of high phosphorus wastewater and environmental management.

**Keywords:** struvite crystallization method; environmental management; ultra-high phosphorus wastewater; basic magnesium carbonate; phosphorus recovery

## 1. Introduction

Nowadays, with the increasing demand for aluminum and its alloys in the world, the surface treatment processes, such as phosphating, are often used to protect the surface of materials from oxidation and corrosion [1,2]. The phosphating process contains a series of chemical reactions, and thus lots of acidic ultra-high phosphorous wastewater was produced [3]. The composition of phosphating wastewater produced in the processing of aluminum profiles is complex, and direct discharge will seriously pollute the surrounding ecological environment [4]. When the total phosphorus (TP) concentration in the water body is higher than 0.02 mg/L, it is considered as eutrophic water, which would cause strong algae metabolism, consume a large amount of dissolved oxygen, cause a large number of aquatic organisms to die, and release a large amount of toxic gases: $H_2S$, $NH_3$, $CH_4$, etc., leading to deterioration of water quality, disrupting the ecological balance and stability of water bodies, and increasing the difficulty and cost of water treatment [5]. So, the removal of phosphorus from wastewater is of great significance for environmental protection.

Current methods for phosphorus removal include biological method, physical adsorption method, electrodialysis method, ion exchange method, chemical precipitation method, etc., [6]. The biological

method is susceptible to factors such as wastewater chemical composition, process parameters, and structure size [7]. Therefore, the process of biological method is unstable, and the phosphorus removal in the form of sludge will eventually cause secondary pollution [8]. The physical adsorption has a disadvantage of low adsorption efficiency of the adsorbent [9]. The electrodialysis process has high requirements on the ion selectivity of the permeation membrane itself, and is only suitable for enterprises with a small amount of wastewater with a single component [10]. Ion exchange method also has high selectivity for exchange objects, and can only handle solutions with lower ion concentrations [11]. The reproducibility of ion exchange resins is also a major problem. The chemical precipitation treatment of phosphorus is to separate the phosphorus by the precipitation reaction between orthophosphate and metal ions. Common metal salts that can react with soluble phosphate to form insoluble matter include calcium, aluminum, ammonium, magnesium salts, and iron [12]. The chemical precipitation method has a high removal rate and is widely used in the treatment of industrial high-concentration phosphorus-containing wastewater, but this method can only "remove phosphorus," not "recycle phosphorus" [13]. Phosphorus is a precious natural resource on the earth, and it is also one of the indispensable elements that make up living matter. The large-scale exploitation of phosphate rock and the neglect of phosphorus resource recovery have led to the depletion of phosphorus resources [14]. Because of the non-renewable nature of phosphorus resources, the sustainable use of phosphorus resources has received increasing attention in recent years [15]. Research on recovering phosphorus using struvite ($MgNH_4PO_4 \cdot 6H_2O$, referred to as MAP) crystallization has also received widespread attention from scientists [16]. Struvite crystallization means that when the concentrations of $Mg^{2+}$, $PO_4^{3-}$, and $NH_4^+$ in wastewater reaches the solubility product ($K_{sp}$) of struvite, $NH_4^+$ and $PO_4^{3-}$ are precipitated and removed as struvite crystals, as shown in formula (1).

$$Mg^{2+} + NH_4^+ + H_nPO_4^{3-n} + 6H_2O \rightarrow MgNH_4PO_4 \cdot 6H_2O\downarrow + nH^+ \tag{1}$$

This method can not only efficiently remove high-concentration ammonia nitrogen and water-soluble phosphorus in wastewater, but also produce struvite as a slow-release fertilizer for agricultural production and flower cultivation [17]. Struvite crystallization has been extensively studied for removal of nitrogen and phosphorus in wastewater [17,18]. However, for high-concentration acidic phosphorus-containing wastewater (total phosphorus content of ~24,500 mg/L) generated in the processing of aluminum profiles, reducing the phosphorus content to the discharge standard (≤5 mg/L) remains a challenge. Thus, the main aim of this work is studying the factors affecting the recovery of phosphorus in acidic ultrahigh-phosphorus wastewater by struvite crystallization, such as the proportion of reactant materials, initial pH, and reaction time. The study found that using basic magnesium carbonate as the source of magnesium for struvite precipitation can take advantage of the alkaline nature of the basic magnesium carbonate aqueous solution, raise the pH of wastewater, and reduce the use of alkali. It can also ensure that the TP content in the treated wastewater meets the standard discharge and recover higher purity phosphorus resources, which has a high potential economic value.

## 2. Materials and Methods

### 2.1. Materials

Calcium oxide was purchased from Tianjin Bodi Chemical Co., Ltd., and magnesium chloride was purchased from Kunming Huangbao Trading Company. Ammonium chloride, calcium chloride, sulfuric acid, magnesium hydroxide, potassium antimony tartrate, nitric acid, and perchloric acid were purchased from Sinopharm Chemical Reagent Co., Ltd. (Shanghai, China). Ammonium molybdate was purchased from Tianjin Chemical Reagent Kaida Plant, and ascorbic acid was purchased from Tianjin Dingshengxin Chemical Co., Ltd.; sodium hydroxide was purchased from Shanghai Titan Technology Co., Ltd. Ammonium bicarbonate was purchased from Shanghai Aladdin Biochemical Technology Co., Ltd. The filter paper (Φ15, medium speed) was purchased from Wendong Chemical

Co., Ltd. (Shanghai, China), and the needle filter (0.22 µm) was purchased from ALWSCI (Hangzhou, China). All reagents were of analytical purity and not further processed before use.

## 2.2. Methods

The wastewater in the experiment is provided by Haiyan County Saixin Metal Surface Treatment Co., Ltd. (Haiyan, China), with the main pollutants in wastewater is shown in Table 1. It was determined that the proportion of phosphorus in soluble orthophosphate to TP was 98.52%. In order to eliminate the influence of sulfate ion in the waste liquid and recover the high-purity phosphorus resources, the waste liquid needs to be pre-treated as follows: an appropriate amount of the above-mentioned wastewater was taken, and calcium oxide was added thereto at a concentration of 38 g/L to remove sulfate. The filter was used to filter out the precipitate. After analysis, it was found that the amount of dry mud produced in this step was about 73 g/L, and the main component was calcium sulfate. The removal rate of sulfate in the waste liquid reached 90%, while the concentration of phosphate was basically unchanged, and the pH of the solution increased from 0.6 to 1.8. Continuously adding sodium hydroxide to the filtered waste liquid to a pH of 4.0, a precipitate is formed due to the presence of small amount aluminum ions in the wastewater. The supernatant was filtered and the magnesium salt and ammonium salt were added according to the set molar ratio. Struvite precipitated in the waste liquid after stirring for 30 minutes, and then it was allowed to stand for 20 minutes. After that, the supernatant was filtered and the TP content was measured.

**Table 1.** The parameters of the wastewater sample.

| Items | Units | Parameters |
|-------|-------|------------|
| pH | | 0.6 |
| TP | mg/L | 24,000–25,000 (avg.: 24,500.00) |
| Al | mg/L | 4300.00 |
| $SO_4$ | mg/L | 50,000–60,000 (avg.: 55,000.00) |
| $NO_3$ | mg/L | 302.90 |
| Cl | mg/L | 164.50 |
| Mo | mg/L | 46.30 |
| Ti | mg/L | 42.10 |
| Fe | mg/L | 77.60 |
| As | µg/L | 17.70 |
| Sb | µg/L | 56.90 |

## 2.3. Analysis

The precipitate was dried at 75 °C to constant weight and ground to a powder to analyze the crystal composition and microstructure by X-ray diffraction (XRD, Rigaku-D/max 2550 PC, Japan) and scanning electron microscope (SEM, Hitachi S-4800, Japan). TP is measured using ammonium molybdate spectrophotometry (Chinese National Standard: GB/T 11894-1989). The model of the spectrophotometer is V1800, Unico Instrument Co., Ltd., Shanghai. The concentration of sulfate ion was measured by ion chromatography (ICS-2100, Dion (China) Co., Ltd.). The concentration of metals ions was measured by inductively coupled plasma atomic emission spectroscopy (ICP-AES, A-6300, Thermo Fisher, USA). The ammonia nitrogen content in the wastewater was measured using the Nessler's reagent spectrophotometry (China Environmental Industry Standard: HJ 535-2009), and the model of the spectrophotometer was the same as that used to measure the TP content.

## 3. Results and Discussion

### 3.1. Effect of Initial pH

The original pH of the experimental wastewater of 0.6 is not suitable for struvite precipitation [19]. If the pH is directly adjusted to be alkaline, the amount of alkaline solution consumed is too large

and the cost is too high. The group of reagents of $Mg_5(CO_3)_4(OH)_2 \cdot 4H_2O$ and $NH_4Cl$ was selected in consideration of the fact that $Mg_5(CO_3)_4(OH)_2 \cdot 4H_2O$ can be dissolved in dilute acid and neutralize the acid in solution to a greater extent than the ammonium radical hydrolysis to produce hydrogen. So adjust the pH of the original wastewater to 4.0, 5.0, and 6.0, and take the filtrate after filtration. According to the concentration of TP in the filtrate, add $Mg_5(CO_3)_4(OH)_2 \cdot 4H_2O$ and $NH_4Cl$ according to the theoretical Mg:N:P molar ratio of 1:1:1. The reaction was stirred for 30 minutes, left to stand for 20 minutes, filtered, the TP content and pH in the filtrate were measured, and the TP removal rate was calculated. The experimental data are shown in Table 2.

**Table 2.** Concentration of total phosphorus (TP) after reaction at different initial pH.

| Initial pH | Mg:N:P | TP (mg/L) | pH After Reaction |
|:---:|:---:|:---:|:---:|
| 4.0 | 1:1:1 | 5.86 | 7.30 |
| 5.0 | 1:1:1 | 6.10 | 7.80 |
| 6.0 | 1:1:1 | 5.68 | 8.10 |

As shown in Table 2, $Mg_5(CO_3)_4(OH)_2 \cdot 4H_2O$ and $NH_4Cl$ were added with Mg:N:P at a molar ratio of 1:1:1. Under different pH conditions, all the TP concentrations were reduced to less than 10 mg/L, and the TP removal rate all reached 99.98%. The pH value of the solution after the reaction is in the range of 7.3–8.2, which belongs to the suitable pH range for struvite crystals. From the data in the table above, it can be seen that the initial pH is suitable between 4 and 6, but it is also necessary to determine the optimal working conditions in combination with the material addition ratio.

*3.2. Effects of Mg/P and N/P Molar Ratios on TP Removal*

Under the condition of N:P as 1:1, adjust the pH to 4.0, 5.0, and 6.0, respectively, add $Mg_5(CO_3)_4(OH)_2 \cdot 4H_2O$ with different Mg: P molar ratio, stir the solution for 30 min and let it stand for 20 min. The effects of Mg:P with different molar ratios on TP removal is shown in Figure 1A. The TP removal in wastewater is improved with increasing of Mg:P molar ratio. Among them, when the initial pH is 4.0, the overall TP removal effect is the best. When Mg:P reached 1.1:1, the TP concentration decreased to less than 5 mg/L. When Mg:P is higher than 1.3:1, the effect of TP removal is basically unchanged or even reduced.

Fixing Mg:P to 1:1, adjusting different pH to 4.0, 5.0, and 6.0, and then $NH_4Cl$ with different N:P molar ratio was added into the wastewater. The effect of molar ratios of N: P on TP removal is shown in Figure 1B. It can be seen that when N:P molar ratio is increased, the TP concentrations in the wastewater also decrease. When the initial pH is 6.0, the overall removal effect of TP is the best, followed by the phosphorus removal effect with the initial pH at 4.0. When N:P is 1.2:1, the minimum remaining amount of TP is 3.74 mg/L. When N: P is higher than 1.2:1, the change of TP concentration is small and tends to be stable.

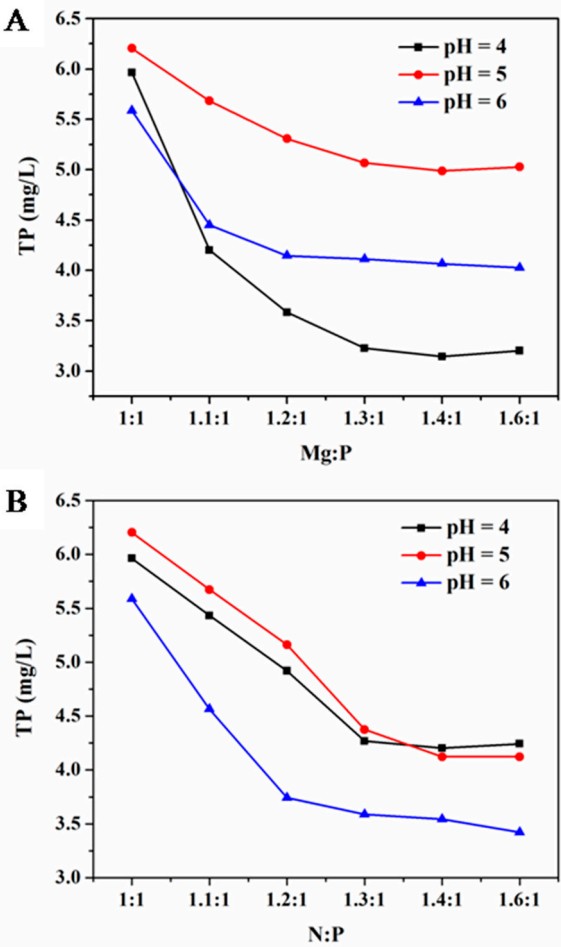

**Figure 1.** Effect of changing Mg:P (**A**) and N:P (**B**) on TP removal under different pH conditions.

### 3.3. Determination of the Optimal Molar Ratio of Mg/N/P

Increasing Mg:P and N:P separately can promote the removal of phosphorus in wastewater, so a set of cross-contrast experiments are designed, as shown in Table 2. The experimental conditions are: adjusting the pH to 4.0, stirring the solution 30 min, and letting it to stand for 20 min. The experimental design and results are shown in Table 3.

**Table 3.** Combination experiment of different Mg:N:P.

| Mg:N:P | TP (mg/L) | TP Removal Rate | pH After Reaction | Sludge Yield (g/L) |
|---|---|---|---|---|
| 1.1:1.1:1 | 3.94 | 99.98% | 7.7 | 43.33 |
| 1.15:1.1:1 | 3.70 | 99.98% | 7.8 | 43.95 |
| 1.2:1.1:1 | 2.98 | 99.99% | 7.8 | 44.70 |
| 1.1:1.15:1 | 3.66 | 99.99% | 7.7 | 43.24 |
| 1.15:1.15:1 | 2.94 | 99.99% | 7.7 | 45.90 |
| 1.2:1.15:1 | 2.86 | 99.99% | 7.7 | 48.86 |
| 1.1:1.2:1 | 3.30 | 99.99% | 7.6 | 43.28 |
| 1.15:1.2:1 | 2.86 | 99.99% | 7.7 | 46.49 |
| 1.2:1.2:1 | 2.78 | 99.99% | 7.7 | 46.86 |

As shown in Table 3, it can be seen that the combined experimental results are better than the individual experimental results. The remaining amount of TP is stable below 5 mg/L, and the pH of the solution after the reaction changes in a small range between 7.6 and 7.9. The sludge yields listed in

Table 3 refer to the mass of the sediment produced in the single process of struvite precipitation after drying. Considering that the increase of NH$_4$Cl dosage will lead to the increase of ammonia nitrogen concentration in the sample, so Mg:N:P = 1.2:1.1:1 is selected as the best dosage ratio, while the TP concentration was 2.98 mg/L.

### 3.4. Effect of Reaction Time on TP Removal

Adjust the pH = 4.0, control the material dosing ratio of Mg:N:P = 1.2:1.1:1, and leave it for 20 min to observe the effect of TP removal under different stirring reaction times. As can be seen from Figure 2, when the reaction time was 10 minutes, the TP concentration was 3.30 mg/L, and it remained almost unchanged when the reaction time reached 20 minutes. It can be seen that the formation rate of struvite crystals is very fast, as observed during the experiment that a large amount of precipitation will occur immediately after the magnesium salt and the ammonium salt are added. Comprehensive consideration, the reaction time of 30 min was selected as the optimal reaction time. At this time, the TP concentration was reduced to 2.98 mg/L, which meet the discharged sewage standard in China (≤5 mg/L, Quality Standards for Sewage Discharge into Urban Sewers, GB 31962-2015) and can ensure the full progress of the reaction and reduce energy consumption.

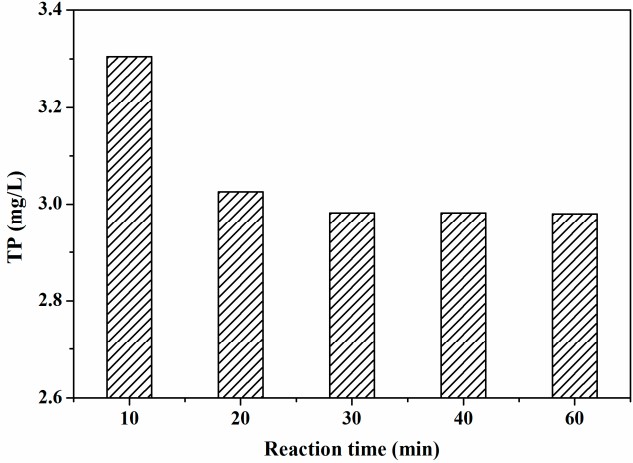

**Figure 2.** Effect of reaction time on TP removal.

### 3.5. Comparison of Struvite Methods Using Different Combinations of Agents

Comparison of struvite methods using three different combinations of agents are listed in Table 4. The TP removal rate by the agents groups of [Mg(OH)$_2$+NH$_4$Cl], [MgCl$_2$·6H$_2$O+NH$_4$HCO$_4$], and [Mg$_5$(CO$_3$)$_4$(OH)$_2$·4H$_2$O+NH$_4$Cl] are 4.91 mg/L, 5.72 mg/L, and 2.98 mg/L, separately. In the three groups of reagents, the consumption of magnesium hydroxide was less, because of its lower solubility [20]. Magnesium chloride has low toxicity and excessive intake is harmful to human health, so it is not suitable for large-scale use to reduce phosphorus ions [21,22]. In terms of the pH value of the solution after the reaction, the pH values of the three groups all meet the discharged sewage standard in China (range of 6.5–9.5, Quality Standards for Sewage Discharge into Urban Sewers, GB 31962-2015). In addition, it can be found that the concentration of sulfate in the wastewater is also reduced. This is because the sulfate and the magnesium salt also react to form a magnesium sulfate precipitate, which can be confirmed from the XRD patterns of struvite precipitation. It can be seen in Figure 3 that the main component of struvite precipitation is magnesium ammonium phosphate, but there is also a small amount of magnesium sulfate [23].

**Table 4.** Comprehensive comparison of ammonium magnesium combined sedimentation agent.

| Parameters | | $Mg(OH)_2 +$ $NH_4Cl$ | $MgCl_2 \cdot 6H_2O +$ $NH_4HCO_4$ | $Mg_5(CO_3)_4(OH)_2 \cdot 4H_2O$ $+ NH_4Cl$ |
|---|---|---|---|---|
| Amount of reagent used (optimal molar ratio in single experiment) | CaO g/L | 38.00 | 38.00 | 38.00 |
| | NaOH mg/L | 14.60 | 28.10 | 14.60 |
| | $Mg(OH)_2$ g/L | 25.87 | / | / |
| | $NH_4Cl$ g/L | 14.84 | / | 16.30 |
| | $Mg_5(CO_3)_4(OH)_2 \cdot 4H_2O$ g/L | / | / | 32.20 |
| | $MgCl_2 \cdot 6H_2O$ g/L | / | 50.85 | / |
| | $NH_4HCO_4$ g/L | / | 25.69 | / |
| Content of residual ions | TP mg/L | 4.91 | 5.72 | 2.98 |
| | $SO_4^{2-}$ mg/L | 5054.00 | 4205.00 | 4415.00 |
| Amount of precipitation | Recycled struvite | 48.60 | 49.80 | 44.70 |
| | CaO pretreatment | 73.00 | 73.00 | 73.00 |
| | Adjust pH | 35.00 | 41.40 | 35.00 |
| pH after reaction | | 8.6 | 7.6 | 7.8 |

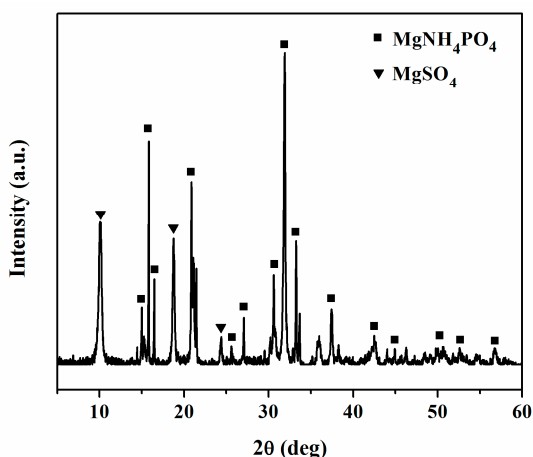

**Figure 3.** XRD patterns of struvite precipitation obtained by the reaction of $Mg_5(CO_3)_4(OH)_2 \cdot 4H_2O$ and $NH_4Cl$.

*3.6. SEM Analysis of Struvite Precipitation*

As shown in Figure 4, the precipitated powder obtained by the reaction of $Mg_5(CO_3)_4(OH)_2 \cdot 4H_2O$ and $NH_4Cl$ was magnified 1500 (A) and 8000 (B) times, respectively. As shown in Figure 4A, the struvite precipitation was observed as guano-like clusters. Combined with the experimental phenomena recorded during the experimental process, it was verified that the struvite deposits formed by the reaction are easy to settle in water, which is beneficial to the recycling and actual use. As shown in Figure 4B, the surface of the struvite sediment can be seen as a whole in a layered form, which indicates that the struvite precipitates in a layered manner, and is arranged neatly and regularly. However, it can also be seen that a small amount of irregular substances are mixed in, and the impurities do not have a specific form. According to the ionic data of the reaction process, it is likely that the residual $Mg_5(CO_3)_4(OH)_2 \cdot 4H_2O$ is undissolved solid, but the proportion is very small.

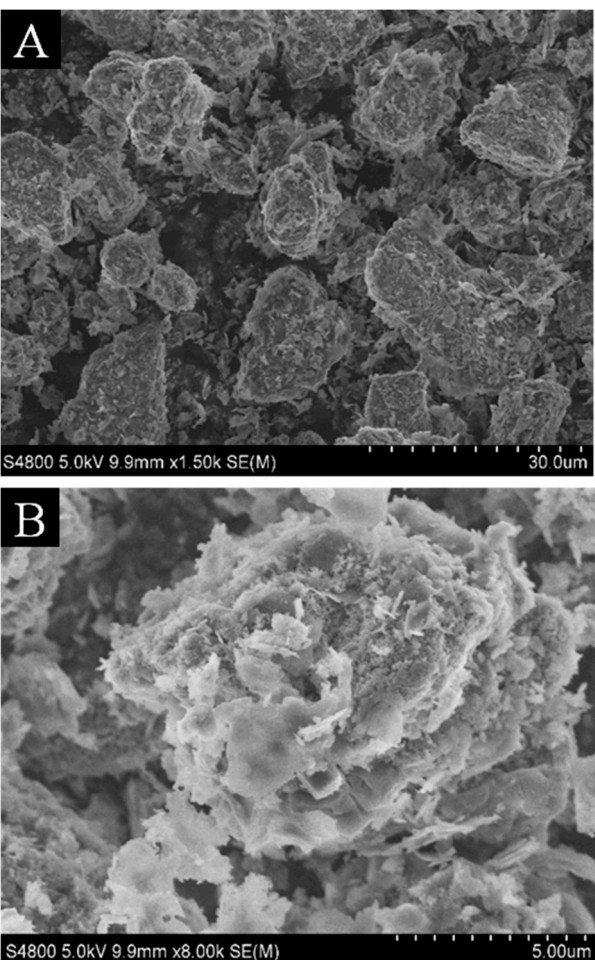

**Figure 4.** SEM diagram of the precipitated powder obtained by the reaction of $Mg_5(CO_3)_4(OH)_2 \cdot 4H_2O$ and $NH_4Cl$ magnified 1500 (**A**) and 8000 (**B**) times.

*3.7. Analysis of Nutrients in Struvite Precipitation*

The nutrient element analysis was performed on the precipitate generated under the conditions of initial pH of 4.0, Mg:N:P of 1.2:1.1:1, stirring reaction time of 30 minutes. The specific operation is: accurately weigh 0.50 g of the dried precipitate, dissolve it in dilute hydrochloric acid, and make it to a volume of 100 mL; during measuring, the concentration is diluted according to different measurement ranges of Mg, N, and P. The mass fraction of Mg, N, and P in the precipitate was calculated based on MgO, N, and $P_2O_5$ [24]. The results are shown in Table 5:

**Table 5.** Composition analysis of struvite precipitation.

| Ingredient Content | $\omega(MgO)$/% | $\omega(N)$/% | $\omega(P_2O_5)$/% |
|---|---|---|---|
| Theoretical value of struvite | 16.43 | 5.70 | 28.92 |
| Actual value of precipitate | 17.08 | 4.00 | 25.22 |

It can be obtained from Table 5 that the reference mass fractions of Mg, N, and P in the precipitate are close to the theoretical values of the corresponding mass fraction in struvite, which indicates that the main constituent of the precipitate is MAP. According to the classification standards of China's phosphate rock, phosphate rock has a mass fraction of $P_2O_5$ in the range of 20% to 30%, which belongs to Grade II phosphate rock. Compared with natural phosphate rock, there is less toxic and harmful

impurities in the struvite precipitation produced in this study, which is easy to purify and separate, and has more economic value [25].

## 4. Conclusions

In summary, the main aim of this work is studying the factors affecting the recovery of phosphorus in acidic ultrahigh-phosphorus wastewater by struvite crystallization, such as the proportion of reactant materials, initial pH, and reaction time. A feasible TP treatment process is designed to make the TP concentration reach the discharge standard in the experiments. This work innovatively uses basic magnesium carbonate as the magnesium source in the struvite precipitation method, which has a higher solubility than magnesium hydroxide, and is more suitable for treating high-concentration acidic phosphorus-containing wastewater. With the conditions of using reagents of [$Mg_5(CO_3)_4(OH)_2 \cdot 4H_2O$ + $NH_4Cl$], initial pH = 4.0, Mg:N:P molar ratio of 1.2:1.1:1, and reaction time of 30 min, the remaining TP concentration was 2.98 mg/L, and the TP removal rate reached 99.99%. XRD pattern analysis confirmed that the main component of struvite sediment was ammonium magnesium phosphate. The nutritional element analysis of the generated struvite precipitated that the mass fraction in $P_2O_5$ reached 25.22%, which is equivalent to high-grade phosphate rock, and it is determined that the recovered struvite has high economic value.

**Author Contributions:** Methodology, Q.L.; validation, S.W. and L.Z.; formal analysis, L.W.; data curation, Q.L.; writing—original draft preparation, Z.S., X.W., and S.W.; writing—review and editing, L.Z.; funding acquisition, L.Z. All authors have read and agreed to the published version of the manuscript.

**Funding:** This research was funded by Natural Science Foundation of China (Nos. 21806101, 51476094, 51590901), Natural Science Foundation of Shanghai (Nos. 16ZR1412600, 15ZR1416900), Gaoyuan Discipline of Shanghai-Environmental Science and Engineering (Resource Recycling Science and Engineering), Shanghai Eastern Professorship grant, Shu Guang project supported by Shanghai Municipal Education Commission and Shanghai Education Development Foundation (No. 15SG52).

**Acknowledgments:** The authors thank Zhongping Xu for his help in SEM characterization.

**Conflicts of Interest:** The authors declare no conflict of interest.

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
