# Peer review of "The Recovery of Phosphorus from Acidic Ultra-High Phosphorous Wastewater by the Struvite Crystallization"

_water, doi:10.3390/w12040946_

Round 1

Reviewer 1 Report

The paper is one of many touching the problem of phosphorus recovery from wastewater by struvite precipitation. The paper is valuable because of the research concerned real wastewater from the aluminum phosphating process.

The authors should clarify the following doubts:

1) The authors focused on real wastewater based from industrial plant. What are the concentrations of metals like e.g: Hg, Cd, Pb, As, Zn, Ti in the wastewater and in the precipitated and  struvite. Do metal ions other than Al, Ca, Mg affect the struvite precipitation?

2) In line 200 authors wrote” magnesium chloride is toxic and harmful to human health” Why? Magnesium chloride not a hazardous substance according to Regulation (EC) No. 1272/2008.

Reviewer 2 Report

The subject of manuscript with title 'The recovery of phosphorus from acidic ultra - high phosphorous wastewater by the struvite crystallization' is very interesting from the environmental point of view.

Some remarks 

Introduction 

What is the main aim of the study? it should be clear defined to close the section of introductory concepts.

Methodological design is clear.

Results

The results are presented clear, figures used to show them are adequate.

In conclusions the attitude to the main aim of study is needed.

Some detail editorial remarks:

  1. In line 84 may be  'treated wastewater' (now is treated water).
  2. Authors report two decimal points in tab. 4 (the values of TP concentration), while in tab. 3 are shown four decimal points. Better choose one format for consistency.

Round 2

Reviewer 2 Report

Thank You for the answer to my suggestion.